# Experience of healthcare personnel on Co-payment mechanism and the implications on its use in private drug outlets in Uganda

**Moses Ocan** [1]*, **Racheal Bakubi**[2], **Loyce Nakalembe**[3], **Deborah Ekusai-Sebatta**[4], **Nsobya Sam**[4]

1 Department of Pharmacology & Therapeutics, College of Health Sciences, Makerere University, Kampala, Uganda, 2 Department of Health Policy, Planning and Management, College of Health Sciences, Makerere University, Kampala, Uganda, 3 Department of Pharmacology, Soroti University, Soroti, Uganda, 4 Infectious Disease Research Collaboration (IDRC), Kampala, Uganda

* ocanmoses@gmail.com

**Data Availability Statement:** All relevant data are within the manuscript and its Supporting Information files.

## Abstract

### Background

Malaria treatment is faced with the challenge of access, affordability, availability, and quality of antimalarial medicines. Affordable medicines facility-malaria (AMFm) program and subsequently Co-payment mechanism were developed to help increase access to quality assured Artemisinin-based combination therapies (ACTs) in seven countries in sub-Saharan Africa. We explored through a qualitative study, experience of healthcare personnel on Co-payment mechanism and the implication on its use in private drug outlets in Uganda.

### Method

Private drug outlets that reported stocking antimalarial agents in moderate-to-high and low malaria transmission settings were purposively selected for inclusion in the study. In each drug outlet, data was collected from a pharmacist/dispenser through key informant interview. The interview was done using a key informant interview guide which covered the following areas, (i) sociodemographic characteristics, ii) awareness of healthcare personnel on the co-payment mechanism, (iii) awareness of healthcare personnel on quality assured artemisinin combination therapies (QAACT), (iv) antimalarial stocking in private drug outlets, (v) antimalarial dispensing prices, (vi) considerations made while stocking, and pricing antimalarial agents, vii) challenges in antimalarial dispensing, and (viii) access to antimalarial agents in private drug outlets. Data was managed using Atlas.ti and analyzed using framework methodology.

### Results

Data was collected from 25 key informants (12 pharmacists and 13 dispensers). Five themes emerged following data analysis, (i) antimalarial stocking influenced by price and client demand, (ii) access and purchasing behavior of drug outlet clients, (iii) basis of dispensing antimalarial agents in private drug outlets, (iv) awareness of QAACT, and (v) awareness

**Funding:** This study is part of the EDCTP2 programme supported by the European Union (TMA2019CDF-2662- Pfkelch13 emergence). The funders had no role in study design, data collection and analysis, decision to publish, or preparation of the manuscript

**Competing interests:** The authors have declared that no competing interests exist

of Co-payment mechanism. None of the study participants was aware of the existence of Co-payment mechanism and QAACT in the private sector. Duocotecin brand of ACTs was the most mentioned and dispensed ACT among the study participants in private drug outlets. Nearly all the pharmacists/dispensers said that many clients who request to purchase ACTs don't come with a prescription and prefer buying cheaper antimalarial agents. Study participants reported stocking and selling both ACTs and non-ACT antimalarial agents in the drug outlets. Pharmacists/dispensers in the drug outlets reported that most clients could not afford buying a full dose of an ACT. None of the study participants considered using Co-payment mechanism while stocking ACTs in the drug outlets.

## Conclusion

There is lack of awareness and utilization of Co-payment mechanism in stocking, pricing, and dispensing of ACTs among pharmacists/dispensers in private drug outlets in Uganda. The antimalarial dispensing in drug outlets was mostly based on prescriptions, clients' preferences, and medicine affordability. The Ministry of Health needs to create demand for Co-payment mechanism through public awareness campaigns, training of healthcare personnel and behavior change communication in the private sector.

## Introduction

In most low-and-middle income countries (LMICs), the private sector is a primary source of healthcare prior to visiting public health facilities [1]. Therefore, efforts to increase availability and access to effective antimalarial agents in the private sector would potentially help improve malaria treatment in communities. The Affordable medicines facility for malaria (AMFm) program was piloted in malaria affected countries in 2010 [2,3]. The AMFm aimed to increase affordability, increase availability, increase use of QAACT, and to stop use of artemisinin monotherapies. Specific interventions included negotiating price reductions with manufacturers, subsidizing cost of ACTs, and supporting interventions to promote appropriate ACT use, such as training and behavior change communication for private sector vendors[4]. Following the end of AMFm pilot period, the Global fund continued at national level in six countries (Ghana, Kenya, Madagascar, Nigeria, Tanzania, and Uganda) to subsidize QAACT in the private sector through Co-Payment Mechanism (CPM) [4]. The CPM leverages Global Fund-negotiated ACT prices and further reduces the price to pharmaceutical importers in participating countries through a co-payment that Global Fund makes directly to manufacturers on their behalf [4,5]. In Uganda, the AMFm was later replaced by Co-payment mechanism in 2015 to ensure continued access to QAACT in the private sector [6]. The CPM focuses exclusively on the private-for-profit sector supply of QAACTs. And in Uganda the government set the subsidy levels at 70% for first-line buyers of quality assured artemisinin-based combination therapies [4].

Artemisinin-based antimalarial agents are the cornerstone of malaria treatment globally [7]. However, unlike previous antimalarial medicines like chloroquine and Sulphadoxine-Pyrimethamine, these agents are relatively more expensive [8]. The high cost of these agents affects access and use in most malaria affected countries globally [8]. Additionally, it is likely to affect the ability to afford full course of malaria treatment among the population [9]. Previous studies have reported high prevalence of use of less than full dose of antimalarial agents in

communities [10]. This is likely to predispose to the risk of unwanted treatment outcomes including treatment failures thus driving resistance development [11,12]. Despite being prescription only medicine, antimalarial agents are accessed over the counter in most LMICs [13]. In Uganda, a study by Ocan et al., [14] reported that a third of the population access ACTs over the counter. This is likely to predispose to the risk of inadequate treatment due to purchasing of incomplete dose because of prohibitive price of artemisinin-based antimalarial agents [9]. The rapid symptom resolution from use of these agents is likely to further predispose to inadequate treatment [15].

In Uganda, malaria treatment is provided without any cost sharing in public health facilities. However, due to frequent drug stock-outs in public health facilities, majority of the population first seek malaria treatment in the private sector [16]. The Co-payment mechanism was an attempt to improve access to effective malaria treatment in the private sector in endemic areas [4]. However, there is no readily available information on the implementation of co-payment mechanism in the private sector in Uganda. In this study, we assessed the experiences of healthcare personnel with Co-payment mechanism and implications on its utilization in stocking, pricing, and dispensing of Antimalarial agents in the private drug outlets.

## Materials and methods

### Ethics statement

The study protocol was reviewed and approved by Makerere University School of Biomedical Sciences Research Ethics Committee (SBS 803). The protocol was further reviewed and cleared by Uganda National Council of Science and Technology (UNCST), (HS1169ES). Administrative clearance was also obtained from the local district authorities in the study districts. In each drug outlet, the data collectors obtained a written informed consent prior to the data collection (key informant interview).

### Study design and setting

An exploratory qualitative study was conducted among healthcare personnel (pharmacist/ drug dispenser) in private drug outlets in moderate-to-high (Apac and Tororo districts) and low (Kabale and Mbarara districts) malaria transmission settings in Uganda. We used the Ministry of health stratification of malaria transmission intensity across the country to identify areas categorized as having moderate-to-high and low malaria transmission. Malaria transmission in the country is stratified into four levels based on malaria cases, 'high' (> 499 cases per 1000 population/year), 'moderate' (200 to 499 cases per 1000 population per year), 'low' level 2 (50 to 199 cases per 1000 population per year), 'very low' level 1 (1–49 cases per 1000 population per year) [17]. Tororo and Apac are districts with moderate-to-high malaria transmission while Mbarara and Kabale are low malaria transmission area [17,18]. The districts were purposely selected following malaria transmission intensity, moderate-to-high (Apac and Tororo) and low (Mbarara and Kabale). Data collection was done from June–December 2021.

### Study population and sampling

The study was done among healthcare personnel (pharmacists/drug dispensers) in private drug outlets in four [4] districts. The districts were purposely selected following malaria transmission intensity, moderate-to-high (Apac and Tororo) and low (Mbarara and Kabale). In each district (Tororo, Apac, Mbarara and Kabale), a comprehensive list of the available private drug outlets was compiled using the National drug authority (NDA) register of drug outlets. Additionally, a census was taken to identify all functional private drug outlets (pharmacies and

drug shops) in each district prior to data collection. From the census and review of drug outlets listed in the NDA register, each of the available drug outlet in the survey districts were considered for inclusion in the survey. All the drug outlets which reported stocking antimalarial agents were purposively selected for inclusion in the study. In each drug outlet, the healthcare personnel (pharmacist/dispenser) were contacted for enrolment into the study. The sample size was arrived at guided by the principle of saturation where the interviews were conducted up to a point where interviews did not generate any new perspectives or information. In each district, all drug outlets in urban settings were visited first followed by those in rural settings.

## Data collection procedure

In each of the included drug outlet, one healthcare personnel (pharmacist/dispenser) were interviewed. The participants were approached face-to-face, and all interviews were conducted at the drug outlets. Data collection was done through key informant interviews (KIIs). The KII guide was developed using information on Co-payment mechanism from previous studies [6,19] (S1 Appendix). The KII guide collected data on the following areas, (i) sociodemographic characteristics, ii) awareness of healthcare personnel on the co-payment mechanism, (iii) awareness of healthcare personnel on quality assured artemisinin combination therapies (QAACT), (iv) antimalarial stocking in private drug outlets, (v) antimalarial dispensing prices, (vi) considerations made while stocking, and price of antimalarial agents, vii) challenges in antimalarial dispensing, and (viii) access to antimalarial agents in private drug outlets. The interview guide was pretested among ten [10] pharmacists in Kampala city. The pretest data was used to adjust the interview guide and clarifying the interview questions. The key informant interviews were conducted in English. Interviews were done by a social scientist with master's degree (OW), a pharmacy technician with diploma in pharmacy and dispenser (KH) and the principal investigator with Doctor of Philosophy degree and lecturer at the department of Pharmacology and Therapeutics, Makerere University (OM). The research assistants were all male and were trained on the study protocol and the interview guide prior to field data collection. The participants did not have any prior knowledge regarding any of the research assistants. For each of the discussion questions, participants were given prompts to guide the interview. The interviews were audio recorded using a voice recorder (SONY, ICD-PX470, China) to ensure all data was captured accurately. Additionally, field notes were taken during each of the interviews.

## Data management and analysis

At the end of each data collection day, audio recordings were transcribed into Microsoft word and additional information from field notes included in the transcripts. A total of 10 transcripts were randomly selected by the principal investigator and compared with the audio recordings to check for accuracy and consistency. To facilitate coding and analysis, the transcribed data were entered into Atlas. ti software *ver* 9.0. [20] Each transcript was given a participant identification number, a code, and a date to maintain anonymity. The analysis followed a framework methodology [21] which is comprised of the following five stages. **Stage one,** familiarization with the data. In this stage, the interview transcripts were reviewed by the analyst to become acquainted with the data and look for ideas, patterns, and developing themes. **Stage two**, development of a coding framework (identifying themes) that is informed by the ideas that emerged from the familiarization stage of the analysis. The code book was developed by two independent team members (RB and MK), and it was reviewed by the principal investigator (OM). **Stage three**, data coding/indexing. This involved identification, organizing and labelling of data into meaningful groups. The developed codes were reviewed by the principal

investigator. **Stage four**, charting and summarizing of coded data. This involved re-organization of coded data to identify emerging themes. **Stage five**, interpretation/mapping of data. In this stage, data was interpreted by identifying key concepts separately and then compiled together to identify any relationships that may exist.

## Results

### Characteristics of study participants

Data was collected among 25 key informants of which 15 (60%) were males and 10 (40%) females. The key informants were pharmacists 12 (48%) and dispensers, 13 (52%). Some respondents [3] each from a separate pharmacy declined to consent to the study. The key informant interviews took on average 45 ± 10 minutes.

### Emerging themes from the analysis

We established the following themes.

1. Antimalarial stocking is influenced by price and client demand

2. Antimalarial access and purchasing behavior of private drug outlet clients.

3. Prescription, affordability, and client preference are the basis of dispensing antimalarial agents in private drug outlets.

4. Awareness of quality assured artemisinin combination therapies (QAACT) and cost of ACTs.

5. Awareness of Co-payment mechanism among healthcare personnel in the private sector

### Antimalarial stocking is influenced by price and client demand

The antimalarial agents commonly mentioned by study participants as being present in their drug outlets included, Duocotexin, Coartem, Fansidar, Quinine and Artesunate. No respondent considered whether an artemisinin-based combination therapy was Quality Assured (QAACT) or not while stocking antimalarial agents. The different considerations that were reportedly followed by pharmacists/dispensers while stocking antimalarial agents in private drug outlets include the following,

a) National guidelines: Study participants reported relying on Uganda clinical guidelines, essential medicines and health supplies list provided by the Ministry of Health while stocking antimalarial agents. Only antimalarial agents listed in the national guidelines were reportedly considered for stocking. First-line ACTs for management of uncomplicated malaria, Artemether-lumefantrine (AL) were the most preferred for stocking in drug outlets than second-line agents like, Dihydroartemisinin-Piperaquine (Duocotexin) and the first-line alternative, Artesunate-Amodiaquine.

*[. . .We try as much as we can to follow the guidelines of the country, so we have the first line drugs which is the artemether lumefantrine, the first line alternative is artesunate and amodiaquine, that is for the uncomplicated cases. Then we have the second line which include, the most common is doucotecxin, then we have injectables, we have artesunate of varying strengths and artemether, then quinine. . .* **Participant_01, High malaria transmission setting]**

*[. . .One is the guidelines, because after investigations, they come up with the type of combinations that are supposed to be used depending on the resistance patterns for the country. . .* **Participant_02, Low malaria transmission setting**]

b) Client demand: Respondents reported aligning their antimalarial stocking with the economic forces of demand and supply. Highly consumed antimalarial medicines or brands were reportedly most preferred by pharmacists/dispensers for stocking in the drug outlets.

*[. . .It is demand. As a community pharmacy, we are driven by demand; that is, what clients want or what clinicians are prescribing is what we stock. So, even the antimalarial medicines we have are few, as you may realize, because we do not receive prescriptions for antimalarial drugs. And we are also driven by the quality of medicines. We stock the ACT brand that we know has been working. That is what basically drives the stocking of antimalarial medicines, the demand, and the quality. . .***Participant_03, Low malaria transmission setting**]

c) Wholesale and retail prices of antimalarial agents: The study participants reported stocking antimalarial medicines with lower wholesale and retail prices. The main reason cited was that majority of the population is poor and cannot afford highly priced medicines.

*[. . .Yes, the first-line treatment I have is cheap, and our local population here is poor, so I have Lumartem brand of Artemether-lumefantrine (AL) which is a bit reasonable and effective. I considered its effectiveness and cheapness, so at least the population can afford it. . .* **Participant_04, High malaria transmission setting**]

d) Study participants (pharmacists/dispensers) reported stocking antimalarial medicines based on malaria burden. The study participants from settings with high malaria burden, reported that stocking antimalarial medicines was based on demand created because of high prevalence of malaria.

*[. . .My motivating factor for stocking these very antimalarial agents that I have in stock is malaria, which is rampant in this district and affects mostly children, and women. So, I felt like let me get the first line treatment for uncomplicated malaria, Artemether-Lumefantrine (AL) instead of the second line (Dihydroartemisinin-Piperaquine) because if I encountered a patient who needed the second line, that one would be beyond me, I would refer. . .* **Participant_05, High malaria transmission setting**]

*[. . .one of the reasons that make us stock antimalarial drugs is because there is a necessity. . . to have antimalarial drugs. . .* **Participant_06, High malaria transmission setting**]

*[. . .we have a high prevalence of malaria, and antimalarial drugs are one of the stock that I have in large quantities because it moves very fast. . .* **Participant_06, High malaria transmission setting**]

### Antimalarial access and purchasing behavior of drug outlet clients

The study participants reported that some of the clients seeking to purchase antimalarial agents had prescriptions while others did not. The pharmacists/dispensers noted that the drug outlet clients who presented prescriptions while purchasing antimalarial agents were those who had first sought treatment from public health facilities. However, when they find that the antimalarial agents are out of stock in the public facilities, they then resort to purchasing antimalarial agents from private drug outlets.

*[. . .When a patient goes to a hospital, they are prescribed antimalarial drugs, but when the drugs are out of stock in the hospital, they come here to the private drug outlets. So, when they come, they have the prescription, and we give them antimalarial medicines according to that prescription. . .* **Participant_07, High malaria transmission setting]**

The study participants reported that clients of private drug outlets mostly preferred to purchase cheaper brands of antimalarial agents. For example, pharmacists/dispensers noted that most clients demand for low priced antimalarial agents like Sulphadoxine-Pyrimethamine (Fansidar), and Artemether-lumefantrine brands like Lumartem.

Drug outlet clients were reportedly not able to purchase a full dose of ACT antimalarial agents. In Kabale district (low malaria transmission setting), the study participants noted that most of the drug outlet clients purchased full dose of ACT antimalarial while in Apac district (high malaria transmission setting) majority were not able to afford a full dose. For the clients who could not afford a full dose of ACTs, the dispensers/pharmacists instituted innovative ways to support such clients pay for their antimalarial medicines. These included,

i) Selling a complete ACT dose to patients in installments: A full ACT dose was divided into fewer tablets depending on how much a client could afford to pay for at a point in time. The pharmacists/dispenser then encouraged such clients to make payments in affordable installments until when they complete paying for the full ACT dose.

*[. . . we don't let them go without the medicine because if they go without, then, that's another problem. So, we give them like a half dose, and these people who are always buying half dose. . .they are nearer here. . . They will be "budgetive" (economical) by the time. . . have swallowed, they have planned for the next half dose. . .* **Participant_08, High malaria transmission setting]**

*[. . .actually, I have people who can buy for UGX 1000 (USD 0.3) now and come in the evening for the full doze. And other people could buy like for UGX 1000 (USD 0.3) now but the next tablets. . . some people can forget now. . .* **Participant_08, High malaria transmission setting]**

*[. . . we don't sell 1 or 2 tablets; we sell 6 tablets if you promise to come back for more. When you buy 6 tablets, you take 4 in the morning and the remaining 2, you will have to come back and buy more 6 for evening and next morning. Before there were people buying only 2 tablets but now, we no longer do that. . .* **Participant_09, High malaria transmission setting]**

ii) Providing ACT antimalarial agents on credit to the drug outlet clients who could not afford to pay for a full dose. This practice was common for clients who were well known to the community drug outlet.

*[. . .because I have a lot of debts with people but because we want them to be alive. . . so if you are to estimate, you find that 50% can afford but then 50% are not able to afford. . .* **Participant_08, High malaria transmission setting]**

*[. . .as I have told you early, they cannot afford but it their right to get full dose, we give them, they bring money later, we don't deny them because we also understand and know that they must take their full dose. . .* **Participant_08, High malaria transmission setting]**

iii) Referring malaria patients who cannot afford to pay for a full ACT dose in the private drug outlet to public (government) health facilities where antimalarial agents are provided free of charge.

*[. . .We refer them to government facilities where they can access antimalarial agents free of charge. We tell them that if they cannot afford it, they should visit a government facility where they can access antimalarial free of charge. . .* **Participant_10, Low malaria transmission setting]**

The pharmacists/dispensers reportedly sold both ACTs and non-ACT antimalarial agents to the clients as per their requests. However, ACTs were reportedly more demanded for by the clients as compared to non-ACTs. The non-ACTs which were frequently requested for by the drug outlet clients included Fansidar and Quinine. The main reasons why drug outlet clients requested to buy non-ACTs for malaria treatment included.

i) Prescriptions by clinicians: the pharmacists/dispensers, noted that most clients who purchased non-ACT antimalarial had prescriptions from clinicians. These were mostly pregnant women who were buying Fansidar (Sulphadoxine-Pyrimethamine) for use in intermittent prophylaxis of malaria in pregnancy (IPT$_p$). And patients who found that antimalarial medicines were out of stock in public facilities.

*[. . .When it comes to pregnant women, we always first advise them to see a doctor because we don't want any problems when it comes to medicating them. They prescribe for them fansidar. . .* **Participant_10, Low malaria transmission setting]**

ii) Clients' preferences: Some drug outlet clients reportedly preferred non-ACTs like Quinine to ACT antimalarial agents. The most common reason for their preference was previous successful experience with use of non-ACTs in treatment of malaria.

*[. . . The clients I give my quinine have prescriptions. However, there are clients who come and say that they have malaria, but what works for them is quinine, and of course, we shall give them what works for them . . .* **Participant_11, High malaria transmission setting]**

*[. . .Actually, I have some people, most of them, for those people who come specifically for what they want, like, ah, Fansidar, some ask, like, "I want Fansidar." There are those who use fansidar alone. Then, there are some few people who use chloroquine alone. . .* **Participant_11, High malaria transmission setting]**

iii) Affordability: The pharmacists/dispensers reported cheaper antimalarial agents. Some of the clients preferred non-ACTs while others requested for low priced ACTs. The pharmacist in one of the drug outlets, mentioned that quinine and Fansidar were the cheapest and most demanded antimalarial agents. The study participants for example reported that, a vial of quinine in the drug outlets is dispensed at a cost of UGX 1000 (USD 0.3), and one strip of Fansidar tablets cost UGX 2000 (USD 0.6) compared to UGX 5000 (USD 1.4) for common Artemether-Lumefantrine (AL) brands like LONART.

*[. . .as I told you before, this is a poor population, so they would prefer coming to buy for UGX 500 (USD 0.14), meaning they cannot afford to buy a dose for one day. They will say, "let me first go and take this one and when I get small money, I will come and buy another one" and yet from my observation, some of them don't come back to complete the dose. . .* **Participant_08, High malaria transmission setting]**

*[. . .Those who can afford a full dose treatment, the population is a bit low, may be 20% just, the rest, I just see them, they come and buy a few, some do come back but others don't come back. . .* **Participant_12, High malaria transmission setting]**

## Prescription, affordability, and client preference are the basis of dispensing antimalarial agents in private drug outlets

In most private drug outlets, dispensing of antimalarial agents was based on prescriptions, clients' preferences, and financial ability of the clients. There was limited reliance on laboratory test results with only a few of the drug outlets reporting use of mRDT on malaria symptomatic clients prior to selling (dispensing) of the antimalarial agents. The respondents commonly dispensed antimalarial agents without confirmation of malaria disease among their clients. The study participants also reported dispensing antimalarial agents based on the presenting symptoms such as fever, loss of appetite, general body weakness, joint pains, nausea, and vomiting.

*[. . . One of the ways we dispense is by prescription and then another way is blind therapy. This is where a patient comes and orders antimalarials on suspicion of malaria because of the symptoms of malaria or when a dispenser in a drug outlet will dispense antimalarial agent depending on the symptom presentation of the clients. . .* **Participant_13, Low malaria transmission setting]**

## Challenges with antimalarial dispensing in private drug outlets

The study participants reported two main challenges commonly encountered while dispensing antimalarial agents. These include,

a) Over the counter purchases: in the drug outlets, the pharmacists/dispensers reported that most of their clients do not present prescriptions when seeking to purchase antimalarial agents. In cases where there are no prescriptions, the antimalarial agents are dispensed based on clinical presentation of the clients.

*[. . .When they come to buy medicine directly, it is difficult to convince them, at times it is difficult, but most of the time when I talk to them, they accept to test and they do the test. It is really a challenge because, at times they resist, when you really insist, they say "I have already tested". So, you find that it is really challenge, meaning, most of them take the drugs without being tested. . .* **Participant_10, Low malaria transmission setting]**

*[. . .Few of them come with prescriptions, but most of them come just to buy over the counter. . .* **Participant_10, Low malaria transmission setting]**

b) Inability of drug outlet clients to purchase a complete ACT dose: most drug outlet clients who seek to purchase Artemisinin-based combination therapies do not have sufficient money to purchase a full dose. The study participants reported that some drug outlet clients especially adults have a tendency of buying ACT doses that are meant for children. In addition, some buy incomplete doses. It was also reported that there was no mechanism of following up clients who purchase ACTs to ensure that they complete their doses and monitoring efficacy of the treatment.

*[. . .The challenge I have is you find an adult requesting to purchase and treat malaria symptoms with ACT antimalarial medicines, but they come with UGX 1000 (USD 0.3) and needs 6 tablets because we have a pediatric ACT dose of 6 tablets, you tell them that this is a pediatric dose, however, they insist to have it and promise to come and buy more tablets. That is the main challenge we face here. So, we end up giving those drugs over the counter. Some of them don't even come back, that means they end up with the pediatric dose because they felt better. . .* **Participant_09, High malaria transmission setting]**

## Awareness of quality assured artemisinin-based combination therapies (QAACT) and cost of ACTs

Nearly all the study participants (pharmacists/dispensers) were not aware of QAACT antimalarial agents in the private sector. A green leaf logo is present in the primary package of all Global Fund subsidized QAACT and is promoted in demand creation activities as an indicator of quality [4]. However, none of the study participants could relate the green leaf logo to quality assured artemisinin-based combination therapies (QAACT).

*[. . .I have not seen physically but I believe that green leaf is telling us that the origin of the drug is the leaf because there are sources of medicines from various items, but this one is specifically from a leaf of a certain tree or from a certain plant which I don't know the name. . .* **Participant_11, High malaria transmission setting]**

*[. . .I do not know; I have not researched about it but maybe it is how it is manufactured. . .* **Participant_12, High malaria transmission setting]**

Most of the study participants reported that drug outlet clients do not indicate any preference for quality assured artemisinin-based combination therapies (ACTs with a green leaf logo) while purchasing antimalarial agents.

The study participants sold QAACT and non-QAACT at relatively similar prices and in some cases QAACT antimalarials were reportedly sold at higher prices.

*[. . . Lumarten with Greenleaf is UGX.4500 (USD 1.3), Then, there is Lariat without Greenleaf is UGX 3500 (USD 0.98). . .* **Participant_13, High malaria transmissions setting]**

*[. . . I have not seen any difference in the buying prices; I order every week; sometimes I receive Greenleaf, and other times I get non-Greenleaf. It may be all combiat, one having Greenleaf and another not having it, but the prices are the same, and that is why our prices also remain the same. . .* **Participant_09, High malaria transmission setting]**

*[. . .Sometimes, the prices of the drugs are given according to the cost price. We do not base it on whether one has a green leaf and the other does not have a green leaf. . .* **Participant_14, Low malaria transmission setting]**

## Awareness of Co-payment mechanism among health personnel in the private sector

All the study participants were not aware and lacked knowledge of co-payment mechanism in the private sector in the country.

*[. . .No, I have never heard of Co-payment mechanism, it is my first time to hear of that. Maybe you can elaborate. . .* **Participant_15, High malaria transmission setting]**

## Discussion

In this study, we found that none of the study participants was aware of the existence of Co-payment mechanism in the private drug sector. In Uganda, ACTs sponsored by the AMFm were first delivered in 2011 with the objective of increasing availability, affordability, and market share of artemisinin-based combination therapies [22]. However, we did not find any

evidence of use of Co-payment mechanism in private drug outlets in the study districts across the country close to a decade since the launch of Co-payment mechanism. This finding is unlike that of a previous study by Freeman *et al.*,[19] in Ghana that reported awareness and use of AMFm in the private sector two years after its introduction. Our study was conducted after over a decade since the launch of AMFm and subsequently the co-payment mechanism and this could contribute to the difference in the findings. However, our finding is reflective of the challenges of implementation and monitoring of government programs especially in low-and-middle income countries. Previous studies have reported on inadequate resource allocation by governments affecting implementation of Co-payment mechanism [23]. The lack of awareness coupled with lack of evidence on use of Co-payment mechanism in private drug outlets found in this study threatens to eradicate the gains achieved from the AMFm pilot program in Uganda [22] and other African countries [19,24]. This is especially the case since private sector is often the first point of care for over 50% of the population in most LMICs [1,14].

The study participants reported presence of both ACTs and non-ACT antimalarial agents in the drug outlets. Duocotexin (ACT), Coartem (ACT), Fansidar (Sulphadoxine-Pyrimethamine, SP), Quinine and Artesunate were mentioned as the most common antimalarial agents in the drug outlets. In Uganda, the use of artemisinin-based combination therapies (ACTs) in malaria treatment was adopted in 2006 following malaria treatment policy change[25]. This was due to high levels of resistance to chloroquine and Sulphadoxine-Pyrimethamine across the country [26]. Chloroquine and SP use was stopped across all malaria endemic countries in sub-Saharan Africa in the late 1990's [27]. However, these antimalarial agents continue to be accessible over the counter in private drug outlets in most sub-Saharan African countries [13]. Communities continue to access both ACTs and non-ACT antimalarial agents over the counter for management of malaria symptoms especially due to their low cost relative to that of the artemisinin-based regimens [14]. This was further confirmed by findings of our current study where drug outlet clients preferred cheaper brands of antimalarial agents. This will likely expose malaria patients to inadequate treatment and potential worsening of disease outcomes as resistance to these agents have persisted in Uganda and across many sub-Saharan African countries [28].

The study participants reported incidences where majority of drug outlet clients could not afford to pay for full dose of ACTs. In many cases the adult clients come with money to buy as few as six [6] tablets of Artemether-Lumefantrine (AL) instead of the recommended adult dose of twenty-four [24] tablets. This is like the findings of a previous study by Talisuna et al., [9] that reported high cost of ACTs as a barrier to effective malaria treatment using artemisinin-based combination therapies (ACTs) in malaria endemic countries. This practice can lead to exposure of malaria parasites to subtherapeutic drug levels which could potentially contribute to treatment failures and resistance development [29]. Artemisinin resistance has recently been reported in northern Uganda [30] and such practices could potentially exacerbate resistance across the country. With no known effective alternative to artemisinin agents in malaria treatment, widespread resistance to ACTs could potentially lead to increase in malaria morbidity and mortality especially in sub-Saharan Africa [31]. There is thus need for governments to establish interventions to ensure appropriate use of ACTs in the population to help mitigate the risk of widespread artemisinin resistance.

The WHO recommends use of artemisinin-based combination therapies (ACT) for treatment of uncomplicated malaria [32]. These drugs are highly effective and have the potential to mitigate development of antimalarial resistance. Unfortunately, despite the WHO recommendation and substantial donor funding, only one in five antimalarial drugs used in malaria treatment in malaria endemic countries are ACTs [33]. Reasons for the low use include high price of ACTs in the private sector where most people seek treatment for fever or suspected malaria

[34]. The price of ACTs is typically more than the price of older, less effective antimalarial drugs such as amodiaquine (AQ) and Sulphadoxine-Pyrimethamine (SP) in the retail sector [35]. We found in our study cases where pharmacists/dispensers reported selling ACTs at higher prices compared to non-ACTs like Fansidar and quinine. This is the case despite existence of government subsidy program, Co-payment mechanism which is intended to improve access to ACTs through reducing the dispensing prices in the private sector in the country. A previous study by Tougher et al.,[24] did not find a substantial price change for ACTs 6–15 months after delivery of subsidized quality assured ACTs to Uganda.

The study had limitations, some potential respondents [3] from the drug outlets declined to participate in the study and therefore their views were not captured. However, we conducted the interviews in other consenting drug outlets until no new information was being got from additional interview (saturation). Therefore, this helped minimize the effect of non-response in some drug outlets. The study did not screen for the possibility of counterfeit 'green leaf logo' on the primary package of the ACTs mentioned by the study participants in the private drug outlets. However, a study by Nayyar, [36] reported a low prevalence of falsified antimalarial agents in sub-Saharan Africa and thus this is unlikely to have affected the findings of our study. The study was conducted in only 4 districts in the country and results may be different in other areas.

## Conclusion

There is lack of awareness and utilization of Co-payment mechanism in stocking, pricing, and dispensing of ACTs among pharmacists/dispensers in private drug outlets in Uganda. Dispensing of antimalarial agents in the drug outlets is mostly driven by the prescriptions, medicine affordability and client preference. Most of the drug outlet clients are not able to purchase a full dose of ACT antimalarial agents. The Ministry of Health needs to create demand for Co-payment mechanism through public campaigns and training of healthcare personnel in the private sector. Furthermore, there is need for regular surveys to monitor implementation of Co-payment mechanism in the country.

## Supporting information

**S1 Appendix. Key informant interview guide.**
(PDF)

**S1 File.**
(PDF)

## Acknowledgments

We acknowledge Mr. Tayebwa Mordecai and Ms. Joanita Birungi for managing and coordinating field data collection. We are grateful to the research assistants, Mr. Olwortho Wilfred, and Mr. Kato Henry for the work done during the field data collection.

## Author Contributions

**Conceptualization:** Moses Ocan, Nsobya Sam.

**Formal analysis:** Racheal Bakubi.

**Funding acquisition:** Moses Ocan.

**Investigation:** Moses Ocan.

**Methodology:** Moses Ocan.

**Software:** Racheal Bakubi.

**Validation:** Loyce Nakalembe, Deborah Ekusai-Sebatta, Nsobya Sam.

**Writing – original draft:** Moses Ocan.

**Writing – review & editing:** Racheal Bakubi, Loyce Nakalembe, Deborah Ekusai-Sebatta, Nsobya Sam.

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
