## [Decision Letter · Decision Letter 0]

7 Feb 2024

PONE-D-24-00298Experience of healthcare personnel on Co-payment mechanism and the implications on its use in private drug outlets in UgandaPLOS ONE

Dear Dr. Ocan,

Thank you for submitting your manuscript to PLOS ONE. After careful consideration, we feel that it has merit but does not fully meet PLOS ONE’s publication criteria as it currently stands. Therefore, we invite you to submit a revised version of the manuscript that addresses the points raised during the review process.

We look forward to receiving your revised manuscript.

Kind regards,

Pyae Linn Aung

Academic Editor

PLOS ONE

Journal Requirements:

2. In the Methods, please provide detailed information about the procedure by which informed consent was obtained from organ/tissue donors or their next of kin. In addition, please provide a blank example of the form used to obtain consent from donors, and an English translation if the original is in a different language.

"This study is part of the EDCTP2 programme supported by the European Union (TMA2019CDF-2662- Pfkelch13 emergence)."

Please state what role the funders took in the study.  If the funders had no role, please state: ""The funders had no role in study design, data collection and analysis, decision to publish, or preparation of the manuscript."" If this statement is not correct you must amend it as needed. 

4. We note that your Data Availability Statement is currently as follows: "All relevant data are within the manuscript and its Supporting Information files."

Reviewers' comments:

Reviewer's Responses to Questions

**Comments to the Author**

1. Is the manuscript technically sound, and do the data support the conclusions?

Reviewer #1: Partly

Reviewer #2: Yes

2. Has the statistical analysis been performed appropriately and rigorously? 

Reviewer #1: N/A

Reviewer #2: N/A

3. Have the authors made all data underlying the findings in their manuscript fully available?

Reviewer #1: No

Reviewer #2: Yes

4. Is the manuscript presented in an intelligible fashion and written in standard English?

Reviewer #1: Yes

Reviewer #2: Yes

5. Review Comments to the Author

Reviewer #1: This study conducted key informant interviews with 25 pharmacists/dispensers in two districts in Uganda to better understand types of anti-malarials stocked and sold, as well as clients’ purchasing behaviors. This study was done in the context of the Global Fund’s Co-payment mechanism which seeks to lower the cost of quality-assured ACTs for first-line buyers of the drug to improve access, particularly given that many people seek care for malaria in the private sector. The study found that most people purchase anti-malarial drugs without a prescription and that many people buy less than a complete dose of ACTs. While the latter is a particularly interesting result, I think the authors need to connect their conclusions more clearly to the evidence to make it more convincing.

Comments:

1. I would have liked more description of the sampling strategy. How were districts chosen? How were high/low malaria transmission defined?

2. Was a listing of all pharmacies selling anti-malarials done in these two districts? If not, how were the pharmacies selected? How was it decided which shops to visit first?

3. I would also have liked more detail on the settings of these pharmacies and about the pharmacies itself.

a. For example, is there a big difference in socio-economic status between the two districts (that might explain why in one district people tended not to buy a complete dose)?

b. How big were the pharmacies? How many people visited them per day? Were they in an urban or rural setting?

4. How long did the interviews take? I suggest including the interview questions in the appendix.

5. I would move the information on refusals into the results section. Is there any information on why they might have refused?

6. The themes could be more descriptive in terms of what actual results were. For example, instead of “Considerations taken while stocking, and pricing of antimalarial agents in private drug outlets”, the authors could say something like “Antimalarial stocking is influenced by price and client demand”

7. Many of the results do not include supporting quotes, which make them hard to evaluate. I think the paper would be stronger if the authors include quotes (or multiple quotes) for all statements. For example:

a. Lines 213 (about the national guidelines)

b. Line 277 (about selling in installments)

c. Line 283 (about providing antimalarials on credit)

d. Line 245 (about stocking antimalarials not being a priority)

8. Some of the statements seem quite strong and it might be helpful to get a sense of how many pharmacists actually reported/mentioned something (a few? some? all?). The authors often say “Study participants reported” or “Pharmacists reportedly sold…” but it’s not clear how many actually said this.

9. The authors say that most drug client outlets who want to buy ACTs can’t afford full dose. Did all the pharmacists report this and how do they know that it’s because of the cost? I did not find the one quote that was included on this (Line 364) particularly convincing as it just mentioned a client buying a pediatric dose.

10. In the abstract it is mentioned that 7/10 clients come without a prescription. Where does this number come from? Did the pharamacists report this?

11. Do the authors have information on the cost of ACTs across the different pharmacies? It might be helpful to report the range of prices for the main types of anti-malarials to verify that ACTs do, in fact, cost more than the other anti-malarials despite the co-payment mechanism.

12. Lastly, it was not clear to me why pharmacists’ knowledge of the copayment mechanism was important. Since it’s a subsidy to first-line buyers, we wouldn’t necessarily expect the pharmacists to know about it, or that it would affect their selling decisions.

Reviewer #2: Title: Experience of healthcare personnel on Co-payment mechanism and the implications on its use in private drug outlets in Uganda.

General comment: A well written manuscript exploring experiences of healthcare personnel on Co-payment mechanism and the implication on its use in private drug outlets in Uganda.

There are a few comments that need to be attended:

Lines 91 & 92: QAACT & ACTs should be first written in full on initial appearance. Rather ACT is subsequently written in full in line 468.

Line 243-252: text and accompanying quote do not tally.

Line 346: statement......about seven in ten clients do not present prescriptions when seeking to purchase antimalarial agents... this phrase appear to present informationas as though it originated from quantitative data. There is need to rephrase it.

6. PLOS authors have the option to publish the peer review history of their article (what does this mean?). If published, this will include your full peer review and any attached files.

Reviewer #1: No

Reviewer #2: No

---

## [Author Response · Author response to Decision Letter 0]

7 Mar 2024

Response to reviewers comments for manuscript PONE-D-24-00298

Thanks for the comments on our manuscript, responding to the comments have helped further improve the manuscript.

Requirment comment #1: 1. Please ensure that your manuscript meets PLOS ONE's style requirements, including those for file naming. The PLOS ONE style templates can be found at 

Response: Thanks this has been rectified in the revised manuscritp.

Requirment comment #2: In the Methods, please provide detailed information about the procedure by which informed consent was obtained from organ/tissue donors or their next of kin. In addition, please provide a blank example of the form used to obtain consent from donors, and an English translation if the original is in a different language.

Response: Thanks for the comment, the statement on procedure for obtaining informed consent has been provided in the revised manuscript. The blank form used for obtaining the consent has also been provided in the revised manuscript. 

Requirment comment #3: Thank you for stating the following financial disclosure: 

"This study is part of the EDCTP2 programme supported by the European Union (TMA2019CDF-2662- Pfkelch13 emergence)."

Please state what role the funders took in the study. If the funders had no role, please state: ""The funders had no role in study design, data collection and analysis, decision to publish, or preparation of the manuscript."" If this statement is not correct you must amend it as needed. 

Response: The statement, "The funders had no role in study design, data collection and analysis, decision to publish, or preparation of the manuscript.” Is correct and has been provided as guided.

Requirment comment #3: We note that your Data Availability Statement is currently as follows: "All relevant data are within the manuscript and its Supporting Information files."

Response: Thanks for the comment, this being a qualitative study we have audio recordings which were analyzed to come up with the current results. Sharing of the audio recordings of participant interviews would violate the ethics approval. The contact of ethics committee is +256784574544. We have however, provided the interview guide that was used in conducting the interviews in the revised manuscript. Additionally, we have attached a sample of transcribed audio recording from one of the key informant interviews. 

Requirement comment #5: When completing the data availability statement of the submission form, you indicated that you will make your data available on acceptance. We strongly recommend all authors decide on a data sharing plan before acceptance, as the process can be lengthy and hold up publication timelines. Please note that, though access restrictions are acceptable now, your entire data will need to be made freely accessible if your manuscript is accepted for publication. This policy applies to all data except where public deposition would breach compliance with the protocol approved by your research ethics board. If you are unable to adhere to our open data policy, please kindly revise your statement to explain your reasoning and we will seek the editor's input on an exemption. Please be assured that, once you have provided your new statement, the assessment of your exemption will not hold up the peer review process.

Response: This was a qualitative study where we audio recorded the respondents and analyzed the audio interviews for this manuscript. A sample of the approved interview guide which was used in the interviews is provided in the revised manuscript. Additionally, we have provided one of the transcription of the audio recordings of the key informant interview. 

Requirement comment #6. Please include your full ethics statement in the ‘Methods’ section of your manuscript file. In your statement, please include the full name of the IRB or ethics committee who approved or waived your study, as well as whether or not you obtained informed written or verbal consent. If consent was waived for your study, please include this information in your statement as well. 

Response: A full ethics statement has been provided in the revised manuscript as guided.

5. Review Comments to the Author

Reviewer #1: This study conducted key informant interviews with 25 pharmacists/dispensers in two districts in Uganda to better understand types of anti-malarials stocked and sold, as well as clients’ purchasing behaviors. This study was done in the context of the Global Fund’s Co-payment mechanism which seeks to lower the cost of quality-assured ACTs for first-line buyers of the drug to improve access, particularly given that many people seek care for malaria in the private sector. The study found that most people purchase anti-malarial drugs without a prescription and that many people buy less than a complete dose of ACTs. While the latter is a particularly interesting result, I think the authors need to connect their conclusions more clearly to the evidence to make it more convincing.

Response: This has been rectified in the revised manuscript. 

Comments:

Reviewer #1 comment 1: I would have liked more description of the sampling strategy. How were districts chosen? How were high/low malaria transmission defined?

Response: We used the Ministry of health stratification of malaria transmission intensity across the country to identify areas categorized as having moderate-to-high and low malaria transmission. Malaria transmission in the country is stratified into four levels based on malaria cases, ‘high’ (> 499 cases per 1000 population/year), ‘moderate’ (200 to 499 cases per 1000 population per year), ‘low’ level 2 (50 to 199 cases per 1000 population per year), ‘very low’ level 1 (1-49 cases per 1000 population per year) (Zalwango et al., 2024). Tororo and Apac are districts with moderate-to-high malaria transmission while Mbarara and Kabale are low malaria transmission area (Nankabirwa et al., 2022; Zalwango et al., 2024). The districts were purposely selected following malaria transmission intensity, moderate-to-high (Apac and Tororo) and low (Mbarara and Kabale). 

Nankabirwa JI, Bousema T, Blanken SL, Rek J, Arinaitwe E, Greenhouse B, et al. Measures of malaria transmission, infection, and disease in an area bordering two districts with and without sustained indoor residual spraying of insecticide in Uganda. PLoS One 2022;17(12):e0279464.

Zalwango MG, Zalwango JF, Kadobera D, Bulage L, Nanziri C, Migisha R, et al. Evaluation of malaria outbreak detection methods, Uganda, 2022. Malar J. 2024;23(1):18.

Reviewer #1 comment 2. Was a listing of all pharmacies selling anti-malarials done in these two districts? If not, how were the pharmacies selected? How was it decided which shops to visit first?

Response: Thanks for the comment, yes listing of all pharmacies selling anti-malarial agents was done in all the four districts. In each district (Tororo, Apac, Mbarara and Kabale), a comprehensive list of the available private drug outlets was compiled using the National drug authority (NDA) register of drug outlets. Additionally, a census was taken to identify all functional private drug outlets (pharmacies and drug shops) in each district prior to data collection. From the census and review of drug outlets listed in the NDA register, each of the available drug outlet in the survey districts were considered for inclusion in the survey. All the drug outlets which reported stocking antimalarial agents were noted and healthcare personnel (Pharmacist/dispenser) contacted for enrolment into the study. In each district, drug outlets in urban settings were visited first followed by those in rural settings. 

Reviewer #1 comment 3. I would also have liked more detail on the settings of these pharmacies and about the pharmacies itself.

a. For example, is there a big difference in socio-economic status between the two districts (that might explain why in one district people tended not to buy a complete dose)?

b. How big were the pharmacies? How many people visited them per day? Were they in an urban or rural setting?

Response: The study was done in four districts, Apac, Tororo, Mbarara and Kabale district. Apac and Tororo have moderate-to-high malaria transmission while Mbarara and Kabale have low malaria transmission. We didn’t explore to establish why malaria patients in some districts were able to purchase full dose while those in other districts were not able. The economic activities in all the four districts is mainly agriculture and farming. We collected data from retail pharmacies in all the four districts. In this study, we didn’t assess how many people visited each pharmacy per day. In each district, data was collected from pharmacies located in both urban and rural settings. 

Reviewer #1 comment 4. How long did the interviews take? I suggest including the interview questions in the appendix.

Response: Thanks for the comment, the interview guide has been provided as supplementary information in the revised manuscript. The interviews took on average 45 minutes. This has been included in the revised manuscript. 

Reviewer #1 comment 5. I would move the information on refusals into the results section. Is there any information on why they might have refused?

Response: Thanks for the comment, we sought consent from participants prior to being interviewed. Since participation in the study was voluntary we didn’t probe for the reasons of refusal to participate in the study. The information on refusals has been also included in the results section. 

Reviewer #1 comment 6. The themes could be more descriptive in terms of what actual results were. For example, instead of “Considerations taken while stocking, and pricing of antimalarial agents in private drug outlets”, the authors could say something like “Antimalarial stocking is influenced by price and client demand”

Response: Thanks for the comment, this has been rectified in the revised manuscript. 

Reviewer #1 comment 7. Many of the results do not include supporting quotes, which make them hard to evaluate. I think the paper would be stronger if the authors include quotes (or multiple quotes) for all statements. For example:

a. Lines 213 (about the national guidelines)

b. Line 277 (about selling in installments)

c. Line 283 (about providing antimalarials on credit)

d. Line 245 (about stocking antimalarials not being a priority)

Response: Thanks for the comment, sorry for this ommission in the initial write up. All the supporting quotes have been incorporated in the revised manuscript. 

Reviewer #1 comment 8. Some of the statements seem quite strong and it might be helpful to get a sense of how many pharmacists actually reported/mentioned something (a few? some? all?). The authors often say “Study participants reported” or “Pharmacists reportedly sold…” but it’s not clear how many actually said this.

Response: Thanks for the comment, this has been corrected throughout in the revised manuscript. 

Reviewer #1 comment 9. The authors say that most drug client outlets who want to buy ACTs can’t afford full dose. Did all the pharmacists report this and how do they know that it’s because of the cost? I did not find the one quote that was included on this (Line 364) particularly convincing as it just mentioned a client buying a pediatric dose.

Response: Thanks for the comment, this was not reported by all the pharmacists. The inability to purchase full dose was common in Apac and Tororo districts areas with moerate-to-high malaria transmission compared to Kabale and Mbarara and is reported in the manuscript. The associated quotes are also provided in the manuscript. 

Reviewer #1 comment 10. In the abstract it is mentioned that 7/10 clients come without a prescription. Where does this number come from? Did the pharamacists report this?

Response: Thanks for the comment, this was reported by the pharmacists during the interview as a rough estimate of the clients who come to purchase ACTs in the pharmacies without a prescritpion on a daily basis. This was provided in the submitted manuscript and has been maintained in the revised manuscript. 

Reviewer #1 comment 11. Do the authors have information on the cost of ACTs across the different pharmacies? It might be helpful to report the range of prices for the main types of anti-malarials to verify that ACTs do, in fact, cost more than the other anti-malarials despite the co-payment mechanism.

Response: This was assessed and reported in a quantitative study that we did seperately and the manuscript is under review. However, for the qualitative study the focus was mainly on the experiences of healthcare personnel with the Copayment mechanism. The respondents, provided general references to the dispensing costs which we did not verify and analyze as part of this study and are thus not reported. 

Reviewer #1 comment 12. Lastly, it was not clear to me why pharmacists’ knowledge of the copayment mechanism was important. Since it’s a subsidy to first-line buyers, we wouldn’t necessarily expect the pharmacists to know about it, or that it would affect their selling decisions.

Response: The study was done among retail pharmacies in communities. In the country there are both subsidized and non-subsidized ACTs in circulation. The pharmacists who run most of the retail pharmacies from time to time stock their pharmacies with various medicines. Their knowledge of Copayment mechanism would help guide the choice of ACTs to stock in the pharmacies. As found from our study, pharmacists reported having fewer subsidized ACTs (QAACT) in stock compared to non-subsidized ACTs with some even having non-ACT antimalarials like chloroquine whose use for malaria treatment was discontinued in the country. This could be due to lack of awareness and knowledge on Copayment mechanism. It is therefore important that the pharmacists in retail drug outlets are aware and knowledgible on Copayment mechanism to help guide stocking and dispensing of ACTs in community pharmacies in the country. 

Reviewer #2:

---

## [Decision Letter · Decision Letter 1]

4 Apr 2024

PONE-D-24-00298R1Experience of healthcare personnel on Co-payment mechanism and the implications on its use in private drug outlets in UgandaPLOS ONE

Dear Dr. Ocan,

Thank you for submitting your manuscript to PLOS ONE. After careful consideration, we feel that it has merit but does not fully meet PLOS ONE’s publication criteria as it currently stands. Therefore, we invite you to submit a revised version of the manuscript that addresses the points raised during the review process.

We look forward to receiving your revised manuscript.

Kind regards,

Pyae Linn Aung

Academic Editor

PLOS ONE

Journal Requirements:

Reviewers' comments:

Reviewer's Responses to Questions

**Comments to the Author**

1. If the authors have adequately addressed your comments raised in a previous round of review and you feel that this manuscript is now acceptable for publication, you may indicate that here to bypass the “Comments to the Author” section, enter your conflict of interest statement in the “Confidential to Editor” section, and submit your "Accept" recommendation.

Reviewer #1: (No Response)

Reviewer #2: All comments have been addressed

2. Is the manuscript technically sound, and do the data support the conclusions?

Reviewer #1: Yes

Reviewer #2: Yes

3. Has the statistical analysis been performed appropriately and rigorously? 

Reviewer #1: N/A

Reviewer #2: N/A

4. Have the authors made all data underlying the findings in their manuscript fully available?

Reviewer #1: No

Reviewer #2: No

5. Is the manuscript presented in an intelligible fashion and written in standard English?

Reviewer #1: Yes

Reviewer #2: Yes

6. Review Comments to the Author

**Reviewer #1:** Thank you for clarifying the sampling procedure and for adding the additional quotes- I believe they have considerably strengthened the paper. I have a few remaining minor comments:

1. I still find the 7/10 number cited in the abstract and in Line 449 very specific (and potentially misleading) for a study based on qualitative interviews. Did multiple pharmacists all give this specific estimate? I suggest removing the number and just saying something like “pharmacists said that many clients don’t come with a prescription.”

2. It would be helpful to note as a limitation that the study was conducted in only 4 districts and results may be different in other areas.

3. Line 203: It would be good to include a citation for the software.

4. I suggest including a note for each participant quote indicating whether they come from a high vs a low transmission district. This might help contextualize the comments a bit.

**Reviewer #2:** I find that the authors have not adequately addressed the data availability statement issue raised in the "requirements".

7. PLOS authors have the option to publish the peer review history of their article (what does this mean?). If published, this will include your full peer review and any attached files.

Reviewer #1: No

Reviewer #2: No

---

## [Author Response · Author response to Decision Letter 1]

10 Apr 2024

Response to Reviewer’s comments on manuscript: PONE-D-24-00298R1

We are grateful for the comments, addressing the comments have helped further improve the manuscript.

Comment #1, Reviewer #1: Thank you for clarifying the sampling procedure and for adding the additional quotes- I believe they have considerably strengthened the paper. I have a few remaining minor comments:

Response: Thanks for the comment

Comment #2, Reviewer #1: . I still find the 7/10 number cited in the abstract and in Line 449 very specific (and potentially misleading) for a study based on qualitative interviews. Did multiple pharmacists all give this specific estimate? I suggest removing the number and just saying something like “pharmacists said that many clients don’t come with a prescription.”

Response: Thanks for the comment, this has been corrected all through the revised manuscript as guided to, “pharmacists said that many clients don’t come with a prescription.”

Comment #3, Reviewer #1: It would be helpful to note as a limitation that the study was conducted in only 4 districts and results may be different in other areas.

Response: This has been noted, the limitation has been incorporated in the revised manuscript. 

Comment #4, Reviewer #1: Line 203: It would be good to include a citation for the software.

Response: Thanks, the reference for the software has been included in the revised manuscript. 

Comment #5, Reviewer #1: I suggest including a note for each participant quote indicating whether they come from a high vs a low transmission district. This might help contextualize the comments a bit.

Response: Thanks for the comment, this has been included in the revised manuscript. 

Comment #6, Reviewer #2: I find that the authors have not adequately addressed the data availability statement issue raised in the "requirements".

Response: We have provided the key informant guide that was used in the interviews and a transcript from an audio recording of an interview. 

Reference 

ATLAS.ti Scientific Software Development GmbH (2021): ATLAS.ti Windows. V. 9.0.0.214 ATLAS.ti Scientific Software Development GmbH. Windows, Berlin

---

## [Editor Report · Decision Letter 2]

17 Apr 2024

Experience of healthcare personnel on Co-payment mechanism and the implications on its use in private drug outlets in Uganda

PONE-D-24-00298R2

Dear Dr. Ocan,

We’re pleased to inform you that your manuscript has been judged scientifically suitable for publication and will be formally accepted for publication once it meets all outstanding technical requirements.

Kind regards,

Pyae Linn Aung

Academic Editor

PLOS ONE
---

## [Editor Report · Acceptance letter]

8 May 2024

PONE-D-24-00298R2 

PLOS ONE

Dear Dr. Ocan, 

I'm pleased to inform you that your manuscript has been deemed suitable for publication in PLOS ONE. Congratulations! Your manuscript is now being handed over to our production team.

Kind regards, 

on behalf of

Pyae Linn Aung 

Academic Editor

PLOS ONE